# Practice Changes in Checkpoint Inhibitor-Induced Immune-Related Adverse Event Management at a Tertiary Care Center

**DOI:** 10.3390/cancers16020369

**Published:** 2024-01-15

**Authors:** Malek Shatila, Farzin Eshaghi, Austin R. Thomas, Andrew G. Kuang, Jay S. Shah, Brandon Zhao, Sidra Naz, Mianen Sun, Sarah Fayle, Jeff Jin, Ala Abudayyeh, Ajay Sheshadri, Nicolas L. Palaskas, Maria C. Franco-Vega, Maria S. Gaeta, Anusha S. Thomas, Hao Chi Zhang, Yinghong Wang

**Affiliations:** 1Department of Gastroenterology, Hepatology, and Nutrition, The University of Texas MD Anderson Cancer Center, Houston, TX 77030, USA; mshatila@mdanderson.org (M.S.); buzzhao0813@tamu.edu (B.Z.); snaz@mdanderson.org (S.N.); asthomas1@mdanderson.org (A.S.T.); hzhang20@mdanderson.org (H.C.Z.); 2Department of Internal Medicine, Baylor College of Medicine, Houston, TX 77030, USA; farzin.eshaghi@bcm.edu (F.E.); andrewkuang2014@u.northwestern.edu (A.G.K.); jay.shah@bcm.edu (J.S.S.); 3Department of Internal Medicine, The University of Texas Health Science Center, Houston, TX 77030, USA; austin.r.thomas@uth.tmc.edu; 4Division of Internal Medicine, The University of Texas MD Anderson Cancer Center, Houston, TX 77030, USA; msun4@mdanderson.org (M.S.); sfayle@mdanderson.org (S.F.); 5Department of Informative Services, The University of Texas MD Anderson Cancer Center, Houston, TX 77030, USA; sjin@mdanderson.org; 6Department of Nephrology, The University of Texas MD Anderson Cancer Center, Houston, TX 77030, USA; aabudayyeh@mdanderson.org; 7Department of Pulmonary Medicine, The University of Texas MD Anderson Cancer Center, Houston, TX 77030, USA; asheshadri@mdanderson.org; 8Department of Cardiology, The University of Texas MD Anderson Cancer Center, Houston, TX 77030, USA; nlpalaskas@mdanderson.org; 9Department of Hospital Medicine, The University of Texas MD Anderson Cancer Center, Houston, TX 77030, USA; mfranco@mdanderson.org; 10Department of Emergency Medicine, The University of Texas MD Anderson Cancer Center, Houston, TX 77030, USA; sgaeta@mdanderson.org

**Keywords:** immunotherapy, quality improvement, immune-mediated colitis, pneumonitis, carditis

## Abstract

**Simple Summary:**

Immune-related adverse events (irAEs) are becoming an increasingly prevalent and well-studied phenomenon. Little is known about how clinical practice has evolved to incorporate our growing knowledge on the subject. We identified temporal trends in irAE management as well as factors associated with improved outcomes among these patients such as timely specialty consultation. These findings can improve the quality of management algorithms for immune-related adverse events at our institution and may inform policies in other institutions.

**Abstract:**

Understanding of immune-related adverse events (irAEs) has evolved rapidly, and management guidelines are continually updated. We explored temporal changes in checkpoint inhibitor-induced irAE management at a tertiary cancer care center to identify areas for improvement. We conducted a single-center retrospective study of patients who developed a gastrointestinal, pulmonary, renal, or cardiac irAE between July and 1 October in 2019 or 2021. We collected patient demographic and clinical information up to 1 year after toxicity. Endoscopic evaluation and specialty follow-up after discharge for patients with gastrointestinal irAEs declined between the 2019 and 2021 periods. Symptom duration and steroid taper attempts also declined. For pulmonary irAEs, rates of specialty consultation, hospital admission and readmission, and mortality improved in 2021 compared with 2019. Follow-up rates after hospital discharge were consistently low (<50%) in both periods. For cardiac irAEs, consultation with a cardiologist was frequent and prompt in both periods. Outpatient treatment and earlier specialty consultation improved outcomes with gastrointestinal irAEs. Our study exploring irAE practice changes over time identified areas to improve management; specifically, timely specialty consultation was associated with better outcomes for gastrointestinal irAEs. These findings can help improve the quality of management algorithms at our institution and may inform policies in other institutions.

## 1. Introduction

Immune checkpoint inhibitors are an increasingly popular treatment option for a growing number of cancers. Immune checkpoint inhibitors such as anti-programmed death protein-1/ligand-1 (PD-1/PD-L1) and anti-cytotoxic T-lymphocyte associated protein-4 (CTLA-4) work to induce an immune antitumor response by stimulating T-cell activity and proliferation. This nonspecific activation of T cells can give rise to inflammatory toxicities in any organ system in the body, referred to collectively as immune-related adverse events (irAEs). Such irAEs are estimated to occur in 34–76% of patients, with around 5–20% developing severe grades of toxicity (≥3, as determined by the Common Terminology Criteria for Adverse Events 5.0 [CTCAEv5]) depending on the class of immune checkpoint inhibitor used [1,2]. Gastrointestinal and pulmonary irAEs are among the most common manifestations of immunotherapy toxicity, and cardiac and renal irAEs are rarer but serious presentations [3]. Mortality rates have been estimated at 0.3–1.3% with colitis, pneumonitis, carditis, hepatitis, and neurotoxicity among the most fatal toxicities [4]. Early recognition and treatment of irAEs is crucial given the associated morbidity and mortality, as well as the impact on the feasibility of future immune checkpoint inhibition [5].

Guidelines regarding the management and treatment of irAEs are rapidly changing and evolving. Management of irAEs is usually supportive and strongly relies on early recognition and prompt intervention with strategies tailored to the organ system affected [6]. The first step in managing irAEs begins with a thorough evaluation to exclude other possible diagnoses and establish a clinical symptom grade. CTCAEv5 grading is the current standard to stratify patients and guide their treatment plan. Patients with grade 1 irAEs typically receive supportive treatment, whereas those with grade 2 irAEs may additionally receive corticosteroids. The mainstay of treating grade ≥ 2 irAEs is immunosuppression with corticosteroids and frequently immunomodulatory agents such as selective immunosuppressive therapy (SIT; e.g., infliximab, vedolizumab) or intravenous immunoglobulins [6].

Our understanding of irAEs has changed rapidly. The growing body of literature on irAEs possesses immense translational value to inform clinical practice, especially considering that many centers and physicians have limited exposure to these cases. It is therefore essential to investigate how well clinical practice has kept up with evolving guidelines to identify and address any ostensible shortcomings. Three studies so far have shown that the implementation of a dedicated irAE team to improve various management parameters led to significantly better patient outcomes [7,8,9]. With this in mind, the aim of the current study was to evaluate the changes in irAE management in a tertiary care cancer center over a 2-year period and to identify aspects of management that may be associated with better patient outcomes.

## 2. Materials and Methods

### 2.1. Study Design and Patient Selection

The current study was a single-center retrospective chart review of all patients who developed an irAE between 1 July and 1 October in 2019 or 2021. Any patient with a new, clinically confirmed gastrointestinal, pulmonary, renal, or cardiac irAE in these time windows was included. These specific toxicities were chosen because they are either very common (gastrointestinal) or have high mortality (pulmonary, cardiac) or hospitalization (gastrointestinal, renal) rates. Patients were considered to have a confirmed irAE if the diagnosis was made by a specialty team, the symptoms were managed as an irAE, or irAE was suspected and no other diagnosis was likely. Patients with established alternate diagnoses or unlikely diagnoses were not included in the analysis. Patients were identified using a natural language processing model. Other systemic toxicities, while important, were not included because they can for the most part be managed on an outpatient basis with a lower risk for mortality or future complications.

### 2.2. Natural Language Processing Model

Systems analysts at the study hospital worked closely with clinical staff to develop a dictionary and rules-based annotator for immunotoxicity and irAEs. These dictionaries and negations were used to create a custom annotator using IBM Content Analytics (New York, NY, USA), which reviewed clinical notes daily to select potential cases of irAEs. A set of filters and business rules for post-processing were developed with the help of the clinical staff. These filters and rules allowed for cross note validation of the natural language processing model findings and improved the precision of the output. Filters and rules included filtering for nonspecific text patterns, validating treatment using immunotherapeutic agents, separating concurrent and sequential agents, and identifying the first visit for an irAE.

### 2.3. Data Collection

Patients were followed for up to 1 year after diagnosis or until 1 July of the subsequent year, whichever came first. We collected data from patient charts, including patient demographics, oncologic history, irAE management (treatment, specialty consultation, procedural evaluation), and irAE outcome (hospitalization, symptom improvement, symptom recurrence, clinical remission, and all-cause mortality). We also collected information regarding peak CTCAEv5 grade and CTCAEv5 grade at the time of specialty consultation for gastrointestinal and pulmonary irAEs.

### 2.4. IrAE Management

Treatment options varied by organ system involved and were chosen based on current irAE management guidelines [6]. Treatments included loperamide, mesalamine, corticosteroids, and selective immunosuppressant treatment (SIT (e.g., infliximab, vedolizumab) for gastrointestinal toxicities; bronchodilators, corticosteroids, and SIT for pulmonary toxicities; hydration, dialysis, corticosteroids, and SIT for renal toxicities; and heart failure medications (e.g., ACE inhibitors, angiotensin receptor blockers, angiotensin receptor/neprilysin inhibitors, mineralocorticoid receptor antagonists), diuretics, corticosteroids, SIT, immunoglobulins, and plasmapheresis for cardiac toxicities. 

Specialty consultation refers to outpatient consultation with a specialist prior to hospitalization, inpatient consultation with a specialist, or follow-up with a specialist after hospital discharge. Patients who presented to an emergency department or were admitted to a hospital inpatient unit on or within 1 day of presentation were excluded from any analysis involving outpatient consultation or treatment because these patients would not have had the opportunity to receive either. 

Procedures included endoscopy for colitis, bronchoscopy for pneumonitis, and biopsy for nephritis and carditis. 

### 2.5. IrAE Outcomes

Symptom improvement was defined as the patient having a sustained decrease in symptom severity for more than 30 days as described by the oncologist’s or specialist’s progress notes. Clinical remission was defined as complete resolution of the patient’s symptoms by the end of their follow-up period. 

### 2.6. Statistical Analysis

Data were analyzed using SPSS 26.0 (Chicago, IL, USA). Comparisons were made between the 2019 and 2021 cohorts. The distribution of continuous variables was summarized by medians and interquartile ranges. The distribution of categorical variables was summarized in terms of frequencies and percentages. Chi-square and Fisher exact tests were used to evaluate associations between two categorical variables. Mann–Whitney U tests were used to compare continuous variables between two groups. Univariate logistic regression was used to identify relationships between different management parameters and outcomes among patients with gastrointestinal toxicities. For all analyses, *p* < 0.05 was considered statistically significant.

## 3. Results

### 3.1. Overall Number of irAE Cases by Organ System as Baseline

Using a natural language processing model, we were able to identify suspected cases of irAEs in the 2019 and 2021 periods at our institution across the entire year. We found that gastrointestinal irAEs were the most common (571 cases in 2019 and 718 cases in 2021), followed by endocrine irAEs (556 and 638, respectively) and cutaneous irAEs (465 and 506, respectively). Cardiac (116 in 2019 and 155 in 2021) and renal (117 in 2019 and 212 in 2021) irAEs were the third and fifth least frequent, respectively. More details can be found in Appendix A.

### 3.2. Demographics

A total of 181 patients were found to have irAEs in the designated three-month period and included in our analysis as shown in the patient selection flowchart in Figure 1. Most of our patients were white (87.8%) and male (68.5%), with a median age of 67 years (interquartile range 58–73 years). The most common malignancies were melanoma (21.5%), lung cancer (24.3%), and genitourinary (26.0%) cancer, and most patients had stage IV disease (69.6%). PD-1/PD-L1 blockade was the most common form of immunotherapy (56.4% of patients), followed by combination therapy (36.5%) and single-agent CTLA-4 blockade (7.2%). Around two-thirds of patients (62.4%) were alive at the end of the follow-up period. This information is summarized in Table 1, with a detailed summary by organ system available in Appendix A. 

### 3.3. Gastrointestinal irAEs

#### 3.3.1. Comparison of Management in 2019 and 2021

A total of 77 new gastrointestinal irAEs were diagnosed during the periods studied, 39 in 2019 and 38 in 2021, with no significant differences in the severity of diarrhea (*p* = 1.000) or colitis (*p* = 0.492) between years (Appendix A). Our findings suggested lower rates of outpatient consultation with a gastroenterologist in 2021 compared to 2019. We observed a slight decrease in rates of treatment prior to hospitalization and inpatient consultation with a gastroenterologist in 2021. We also saw a higher rate of hospitalization and hospital readmission rates in 2021 compared to 2019. None of these differences achieved statistical significance however (*p* > 0.05). More importantly, there was a significant decline between the 2019 and 2021 periods in rates of follow-up with a gastroenterologist after discharge (92% in 2019 compared with 48% in 2021, *p* = 0.007), endoscopic evaluation (95% compared with 61%, *p* < 0.001), and SIT use (74% compared with 42%, *p* = 0.004). In terms of outcomes, clinical remission and recurrence rates were similar, with significant improvements in symptom duration (median 43.5 days in 2019 compared with 24.5 days in 2021, *p* = 0.036) and the need for more than two steroid tapering courses (15% compared with 0%, *p* = 0.014). More comparisons between the two periods can be found in Table 2, with a breakdown by diarrhea and colitis severity found in Appendix A. A graphical representation of the case load and hospitalization and procedure rates, median durations of symptoms and hospitalization, and times to procedure and follow-up can be found in Appendix A, respectively.

#### 3.3.2. Differences in Management between Patients Who Did and Did Not Receive an Outpatient Consultation with a Gastroenterologist

Of the 50 patients whose gastrointestinal irAE was diagnosed prior to hospitalization, 39 were seen by a gastroenterologist at our hospital, and 11 were not (Appendix A). Patients who were seen by a gastroenterologist had similar rates of outpatient treatment to that of those who were not (87% compared with 64%), as well as similar rates of hospitalization (49% compared with 64%), symptom recurrence (38% compared with 45%), and clinical remission (92% compared with 82%). None of these differences were significant (*p* > 0.05). Patients who were seen by a gastroenterologist were also more likely to receive follow-up after hospital discharge (84% compared with 29%, *p* = 0.007) and more likely to undergo endoscopic evaluation (82% compared with 45%, *p* = 0.023). There was no significant difference in diarrhea or colitis severity between the two groups. 

#### 3.3.3. Univariate Analysis for Relationships between Management Parameters

A summary of the univariate analysis conducted to evaluate associations between different parameters can be found in Appendix A. Outpatient treatment was associated with less recurrence (odds ratio [OR] 0.2, *p* = 0.005), as were SIT use (OR 0.3, *p* = 0.012) and initial symptom improvement within 30 days of diagnosis (OR 0.04, *p* < 0.001). A longer duration of both symptoms (OR 1.02, *p* = 0.018) and initial steroid use (OR 1.02, *p* = 0.025) predicted a need for more than one steroid tapering attempt. Finally, early consultation with a gastroenterologist (up to 2 weeks after initiation of steroid therapy) was associated with a lower likelihood of colitis recurrence (OR 0.4, *p* = 0.047), hospital readmission (OR 0.2, *p* = 0.012), and need for more than one steroid tapering attempt (OR 0.3, *p* = 0.024).

### 3.4. Pulmonary irAE Management in 2019 and 2021

A total of 76 new pulmonary irAEs were diagnosed in the periods studied, with 38 cases per period. Pneumonitis severity significantly differed between the two periods (78.9% of patients had grade 3 and above pneumonitis in 2019 compared with 57.9% in 2021, *p* = 0.048), as shown in Appendix A. Management of pulmonary irAEs had improved in several ways by 2021. Specifically, rates of outpatient consultation with a pulmonologist prior to hospitalization improved from 67% in 2019 to 93% in 2021 (*p* = 0.055), as did rates of hospitalization (95% compared with 68%, *p* = 0.011), hospital readmission (47% compared with 31%, *p* = 0.179), the need for more than one steroid tapering attempt (42% compared with 24%, *p* = 0.140), and all-cause mortality (68% compared with 32%, *p* = 0.003). Rates of follow-up with a pulmonologist after hospital discharge and bronchoscopic evaluation remained low across both periods, hovering just under 50%. Further details can be found in Table 3, with a breakdown by pneumonitis severity found in Appendix A. No subgroup analyses could be carried out owing to small sample sizes, and univariate analysis revealed no significant correlations except for an association between early consultation with a pulmonologist and lower likelihood of hospitalization (OR 0.13, *p* = 0.003). A graphical representation of case load and hospitalization and procedure rates, median durations of symptoms and hospitalization, and times to procedure and follow-up can be found in Appendix A, respectively.

### 3.5. Renal irAE Management in 2019 and 2021

A total of 15 patients developed renal irAEs within the periods studied, 6 in 2019 and 9 in 2021. The main differences between management between the two periods were a shorter median time to consultation with a nephrologist (median 67.5 days in 2019 compared with 3 days in 2021, *p* = 0.059), a higher rate of inpatient consultation with a nephrologist (33% compared with 83%, *p* = 0.226), and a higher rate of SIT use (0% compared with 44%, *p* = 0.103). All other parameters were consistent across the two periods except for hospital readmission rates, which increased in 2021 (0% compared with 50%, *p* = 0.228). A summary of these findings can be found in Table 4. A graphical representation of case load and hospitalization and procedure rates, median durations of symptoms and hospitalization, and times to procedure and follow-up can be found in Appendix A, respectively.

### 3.6. Cardiac irAE Management in 2019 and 2021

A total of 13 patients developed cardiac irAEs within the periods studied, 4 in 2019 and 9 in 2021. All patients who developed cardiac irAEs were hospitalized, with only one patient diagnosed as an outpatient and seen by a cardiologist prior to hospitalization. All patients received inpatient consultation with a cardiologist as well as cardiac imaging. All patients received prompt consultation with a cardiologist, at a median of 1 day after diagnosis in both periods. Management did not differ between the two periods, but there appeared to be a trend towards higher rates of symptom improvement in 2021 compared with 2019 (89% compared with 50%, *p* = 0.203) and lower readmission rates (11% compared with 25%, *p* = 0.202). A summary of these findings can be found in Table 4. A graphical representation of case load and hospitalization and procedure rates, median durations of symptoms and hospitalization, and times to procedure and follow-up can be found in Appendix A, respectively.

## 4. Discussion

Our study is one of the few quality improvement projects exploring the management of irAEs. We studied changes in the management of irAEs between 2019 and 2021 and identified key aspects associated with better outcomes. We found that overall, patient outcomes improved from 2019 to 2021 despite a decline in performance for a few parameters, namely rates of procedural evaluation and post-discharge follow-up. We also noted that timely specialty consultation may be associated with a lower frequency of outcomes such as hospital admission and readmission, disease recurrence, and the need for multiple steroid tapering attempts. Our findings highlight the potential overburdening of specialty services and the importance of implementing treatment algorithms that ensure adequate and prompt patient follow-up.

Immune-mediated diarrhea and colitis is among the most common irAEs and the most frequently severe [3]. Considering the breadth of the literature on the subject, it was surprising to see that rates of endoscopic evaluation and SIT use dropped considerably between 2019 and 2021 when studies have shown that these interventions are associated with positive outcomes [10,11]. Adding to the complexity of this issue is the fact that despite a decline in these key parameters, several patient outcomes such as remission and recurrence rates, symptom duration, and the need for multiple steroid tapering attempts actually improved by 2021. Together, these findings seem to suggest that our management of immune-mediated diarrhea and colitis has improved over the years, but there is potential for further improvement. 

We observed no differences in disease severity between 2019 and 2021. This suggests that any worsening in parameters is related to a difference in practice rather than in clinical presentation. There are many potential explanations for these changes in practice, as described in Figure 2. The impact of the COVID-19 pandemic cannot be overstated. Multiple studies have shown that consultation [12,13] and procedure [14,15] rates dramatically decreased following the start of the pandemic. Other studies stress the impact of the pandemic-related economic recession on health insurance coverage [16]. At a more basic level, the ever-growing number of patients presenting to the hospital with irAEs may lead to overwhelming the specialty consultation system. Regarding SIT use in particular, better management of immune-mediated diarrhea and colitis could have allowed symptoms to resolve before SIT was needed. This is supported by the decreased need for multiple steroid tapering attempts. Another layer to this could be the increasing comfort of primary oncology teams in handling irAEs without specialty consultation. The establishment of a dedicated irAE committee at the study hospital has helped facilitate this by making great efforts to educate primary providers via lectures, developed algorithms, and educational phone applications. 

The increasing comfort of primary oncology teams in handling certain irAEs is a positive development, but it should not detract from the significance of specialty consultation. Our findings show that specialty consultation—especially early in the disease course—may be linked to better patient outcomes either directly or indirectly through an association with other aspects of management that are correlated with outcomes, such as outpatient treatment. This supports findings from a previous quality improvement project showing that implementation of a specialty service dedicated to gastrointestinal irAEs was associated with shorter symptom duration, higher follow-up rates for gastrointestinal irAEs after hospital discharge, fewer hospital readmissions, and less disease recurrence [8]. This previous study additionally found that patients with immune-mediated diarrhea and colitis who followed up with a gastroenterologist had longer overall survival. However, the utility of specialty services to mitigate poor outcomes after irAEs would ideally be studied in a randomized controlled trial comparing usual care to systematic, early specialty service consultation. 

Pulmonary irAEs, although not as common as gastrointestinal irAEs, are very serious and potentially lethal [17,18]. For this reason, prompt and effective management is critical to ensure patient survival. The current study showed that there have been marked improvements in consultation with a pulmonologist both on an outpatient basis prior to hospitalization and on an inpatient basis. This is likely due to an increased institutional effort to host educational events to raise awareness regarding these toxicities. However, our study showed a few shortcomings as well. SIT use decreased from 18% in 2019 to 8% in 2021, which may be due to the lower frequency of high-grade pulmonary irAEs in 2021 compared with 2019. This change in severity likely had broad downstream impacts on other metrics and may have also affected parameters such as hospitalization, readmission, and mortality rates, and therefore, it is not possible to conclude that consultation with a pulmonologist had a causal role in mitigating poor outcomes after pneumonitis. Furthermore, rates of follow-up with a pulmonologist after discharge and bronchoscopy rates were below 50% in both periods studied. Although bronchoscopy is contingent on several factors, including patient safety, ideally follow-up after hospital discharge should be universal. The low rate of follow-up is likely multifactorial, but considering the information in Figure 2, we would have expected a drop in these parameters. The fact that the rates stayed consistent across the two periods studied is reassuring, but further improvements may be beneficial for patients with pneumonitis. 

Although we cannot say for sure whether the improvement in mortality was due to an improvement in pneumonitis management or a difference in pneumonitis severity, this is still a notable improvement for this particular irAE. Immune-mediated pneumonitis, unlike some other irAEs, is associated with mortality; a study of patients presenting to the emergency department with irAEs showed that the presence of pneumonitis was associated with worse overall survival [19,20,21,22]. This is a difficult irAE to manage in the acute care setting given the complexity of the affected population, including a range of underlying cancers, comorbidities, and alternative diagnoses. Given the challenge in diagnosing pneumonitis, increasing the rate of consultation with pulmonologists and up-front diagnostic bronchoscopy is likely to be beneficial, but definitive proof would require comparison to usual care in a randomized controlled trial. 

In a similar vein, immune-mediated myocarditis is a relatively rare irAE with a high reported mortality [23]. Our findings reflect this; all patients who presented with carditis were hospitalized for this condition and received rapid and comprehensive multidisciplinary management efforts: 100% of patients received a consultation with a cardiologist and 90% of patients underwent a biopsy, most within days of presentation. Patients with pulmonary and cardiac irAEs in our analysis had the highest and second-highest mortality rates, respectively.

As a quality improvement project, our study identified key areas that need to be worked on to enhance patient outcomes. The implementation of automatic order sets in particular may help to ensure a timely diagnosis and workup and improve consultation, treatment, and procedure rates. These types of interventions have already shown benefits in the early diagnosis and treatment of various other pathologies, including sepsis, at multiple institutions [24]. The creation of a dedicated irAE group available on demand may also facilitate improved management of irAEs, as demonstrated by the implementation of such a service at multiple cancer centers [7,8,9]. Nonetheless, our study is not without limitations. Our study is retrospective, which may have impacted the accuracy of data collected from electronic health records. Additionally, we may have missed information regarding management that was carried out at local institutions. The power of our statistical analysis may have been limited by our sample size and time reviewed. Reviewing more patients over a longer follow-up period may help delineate clearer trends in patient management and outcomes. It is also difficult to assess the extent of the impact of the COVID-19 pandemic on health care. Finally, our study explored irAEs in only four organ systems. There would be merit in expanding the sample size and including irAEs from other organ systems such as cutaneous or neurological irAEs.

## 5. Conclusions

The current study is one of the few qualitative studies exploring current practice in managing irAEs. The changes in practice between 2019 and 2021 that we focused on likely reflect a variety of factors, including an evolving knowledge of this disease, as well as the impact of the COVID-19 pandemic. Although the overall irAE outcomes among the toxicities that we described have improved in recent years, there is room for more improvement. Ensuring adequate and timely consultation, procedural evaluation, and the use of SIT could help reduce hospital readmission and disease recurrence, though prospective studies are needed to confirm these findings. Together, these findings can help improve the quality of irAE management algorithms at our institution and potentially others.

## Figures and Tables

**Figure 1 cancers-16-00369-f001:**
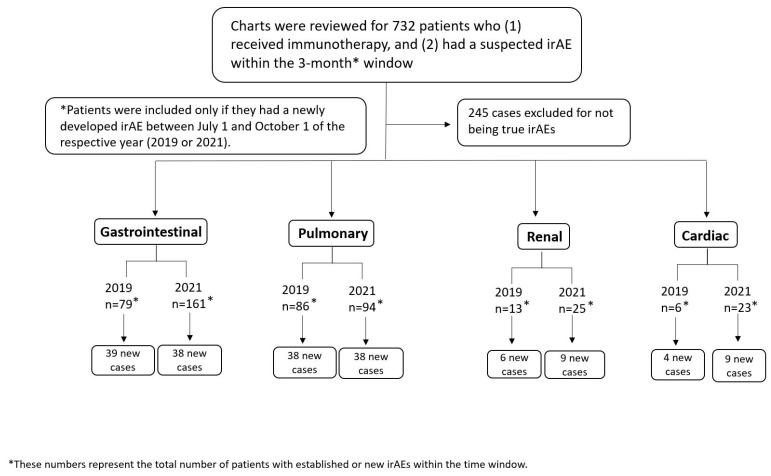
Patient selection flowchart for patients with immune-related adverse events (irAEs).

**Figure 2 cancers-16-00369-f002:**
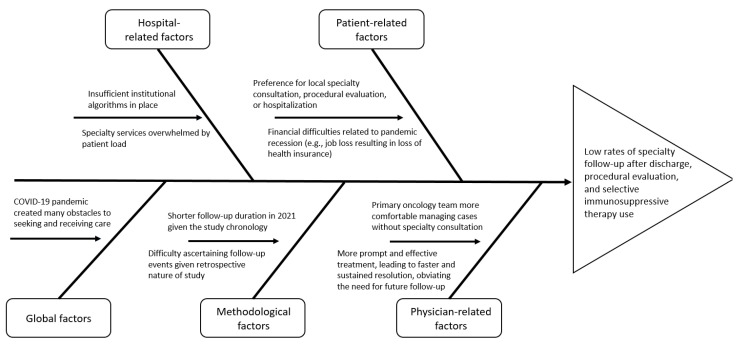
Fishbone diagram exploring reasons for changes in clinical practice.

**Table 1 cancers-16-00369-t001:** Cohort demographic information, n = 181.

Characteristic	No. (%)
Median age (IQR), years	67 (58–73)
Number of infusions (IQR)	4 (2–8)
Median duration of immunotherapy (IQR), days	81 (33–161)
Gender	
Male	124 (68.5)
Female	57 (31.5)
Race	
White	159 (87.8)
Others	22 (12.2)
Malignancy	
Melanoma	39 (21.5)
Genitourinary	47 (26.0)
Lung	44 (24.3)
Gastrointestinal	17 (9.4)
Head and neck	8 (4.4)
Other	26 (14.4)
Cancer stage	
I	4 (2.2)
II	8 (4.4)
III	33 (18.2)
IV	126 (69.6)
Unknown	10 (5.5)
Type of immunotherapy	
CTLA-4 alone	13 (7.2)
PD-1/PD-L1 alone	102 (56.4)
Combination	66 (36.5)
All-cause mortality	68 (37.6)

**Table 2 cancers-16-00369-t002:** Gastrointestinal immune-related adverse-event clinical management and disease outcomes in 2019 and 2021.

Outcome	No. of New Colitis Cases ^1^ (%)		*p*
	2019, n = 39	2021, n = 38	
Median symptom duration (IQR), days	43.5 (18.8–90.8)	24.5 (12.5–46.5)	0.036 *
Outpatient consultation with a gastroenterologist before hospitalization ^2^	23 (82)	16 (73)	0.502
Any consultation with a gastroenterologist ^3^	37 (95)	33 (87)	0.262
Outpatient treatment for gastrointestinal immune-related adverse event before hospitalization ^2^	24 (86)	16 (73)	0.266
Median time to consultation with a gastroenterologist (IQR), days	21 (12–41.5)	16 (5.5–41.5)	0.365
Hospitalization	25 (64)	27 (71)	0.515
Median length of hospital stay (IQR), days ^4^	5 (3–8.75)	7 (3–8)	0.495
Inpatient consultation with a gastroenterologist ^4^	22 (88)	21 (78)	0.267
Follow-up with a gastroenterologist after discharge ^4^	23 (92)	13 (48)	0.007 *
Median time to follow-up with a gastroenterologist (IQR), days ^4^	20 (7.75–58.75)	31 (14–57)	0.451
Endoscopic evaluation	37 (95)	23 (61)	<0.001 *
Median time from immune-related adverse event to endoscopy (IQR), days	7 (2–12.25)	8 (2–34.25)	0.468
Follow-up endoscopy after first endoscopy	14/37 (38)	6/23 (26)	0.408
Symptom improvement	28 (72)	28 (74)	0.411
Median duration of steroid use (IQR), days	35 (28–88)	39.5 (27.75–66.75)	0.880
Need for >1 steroid tapering course	15 (38)	7 (18)	0.06
Need for >2 steroid tapering courses	6 (15)	0 (0)	0.014 *
Selective immunosuppressive therapy ^5^	29 (74)	16 (42)	0.004 *
Response/remission at final follow-up ^6^	36 (92)	34 (89)	0.711
Hospital readmission ^4^	5 (20)	7 (26)	0.743
Recurrence	13 (33)	9 (24)	0.605
All-cause mortality	14 (36)	9 (24)	0.323

* Statistically significant. ^1^ Total cases within the 3-month study window: 2019 = 79; 2021 = 161. ^2^ In total, 50 patients presented with a gastrointestinal immune-related adverse event prior to hospitalization (28 in 2019 and 22 in 2021). The remaining patients all presented at the emergency department or hospital. ^3^ Includes as an outpatient before hospitalization, as an inpatient, or as an outpatient after hospital discharge. ^4^ For those admitted to the hospital, 2019 = 25, 2021 = 27. ^5^ Selective immunosuppressive therapy included infliximab, vedolizumab, or ustekinumab. ^6^ Defined as symptom improvement to Common Terminology Criteria for Adverse Events 5.0 grade 1 or below.

**Table 3 cancers-16-00369-t003:** Pneumonitis clinical management and disease outcomes in 2019 and 2021.

Outcome	No. of New Pneumonitis Cases ^1^ (%)		*p*
	2019, n = 38	2021, n = 38	
Median symptom duration (IQR), days	38.5 (13.0–105.0)	54 (22–70)	0.911
Outpatient consultation with a pulmonologist before hospitalization ^2^	6 (67)	13 (93)	0.055
Any consultation with a pulmonologist ^3^	37 (97)	38 (100)	1.000
Outpatient treatment for pneumonitis before hospitalization ^2^	6 (67)	11 (79)	1.000
Median time to consultation with a pulmonologist (IQR), days	2 (1–13.0)	2 (0–6.5)	0.457
Hospitalization	36 (95)	26 (68)	0.011
Median length of hospital stay (IQR), days ^4^	8 (6–14)	12 (6–13)	0.567
Inpatient consultation with a pulmonologist ^4^	32 (89)	25 (96)	0.388
Intensive care unit admission	15 (39)	10 (26)	0.329
Follow-up with a pulmonologist after discharge ^4^	15 (42)	11 (42)	1.000
Median time to follow-up with a pulmonologist (IQR), days	29 (13.0–98.0)	34 (16–54)	1.000
Bronchoscopic evaluation	19 (50)	18 (47)	1.000
Median time from pneumonitis diagnosis to bronchoscopy (IQR), days	3 (2–17)	3 (2–10)	0.798
Symptom improvement	19 (50)	23 (61)	0.489
Median duration of steroid use (IQR), days	31 (21–56)	35 (22–56)	0.791
Need for >1 steroid tapering course	16 (42)	9 (24)	0.140
Selective immunosuppressive therapy ^5^	7 (18)	3 (8)	0.309
Pneumonitis response/remission at final follow-up ^6^	20 (53)	29 (76)	0.891
Hospital readmission ^4^	17 (47)	8 (31)	0.179
Recurrence	18 (47)	15 (39)	0.644
All-cause mortality ^7^	26 (68)	12 (32)	0.003 *

* Statistically significant. ^1^ Total pneumonitis cases: 2019 = 86, 2021 = 94. ^2^ In total, 23 patients presented with pneumonitis prior to hospitalization (9 in 2019 and 14 in 2021). The remaining patients all presented at the emergency department or hospital. ^3^ Includes as an outpatient before hospitalization, as an inpatient, or as an outpatient after hospital discharge. ^4^ For those admitted to the hospital, 2019 = 36, 2021 = 26. ^5^ Selective immunosuppressive therapy included infliximab, tocilizumab, mycophenolate, or cyclophosphamide. ^6^ Defined as symptom improvement to Common Terminology Criteria for Adverse Events 5.0 grade 1 or below. ^7^ In total, 10 patients (43.5%) in 2019 and 8 patients (66.7%) in 2021 died of pneumonitis or treatment for pneumonitis, *p* = 0.193.

**Table 4 cancers-16-00369-t004:** Nephritis and carditis clinical management and disease outcomes in 2019 and 2021.

Outcome	No. of New Cases ^1^ (%)		*p*
	2019	2021	
Nephritis	n = 6	n = 9	
Median symptom duration (IQR), days	77.0 (28.0–116.0)	19 (14–28)	0.126
Outpatient consultation with a nephrologist before hospitalization ^2^	4 (100)	2 (50)	0.429
Any consultation with a nephrologist ^3^	6 (100)	8 (89)	1.000
Outpatient treatment for nephritis before hospitalization ^2^	3 (75)	3 (75)	1.000
Median time to consultation with a nephrologist (IQR), days	67.5 (34–88)	3 (0–8)	0.059
Hospitalization	3 (50)	6 (67)	0.622
Median length of hospital stay (IQR), days	3 (3–14)	10 (5–11)	0.567
Inpatient consultation with a nephrologist	1/3 (33)	5/6 (83)	0.226
Follow-up with a nephrologist after discharge	1/3 (33)	3/6 (50)	1.000
Time to follow-up with a nephrologist (IQR), days	86	31 (29–181)	1.000
Renal biopsy	3 (50)	4 (44)	1.000
Median time from nephritis diagnosis to biopsy (IQR), days	37 (16–301)	22 (6–51)	0.400
Symptom improvement	5 (83)	6 (67)	0.604
Median duration of steroid use (IQR), days	34 (27–42)	24 (22–35)	0.138
Need for >1 steroid tapering course	2 (33)	3 (33)	1.000
Need for >2 steroid tapering courses	2 (33)	1 (11)	0.545
Selective immunosuppressive therapy ^4^	0 (0)	4 (44)	0.103
Nephritis response/remission at final follow-up ^5^	4 (67)	5 (56)	1.000
Hospital readmission	0 (0)	3/6 (50)	0.228
Recurrence	4 (67)	6 (67)	1.000
Mortality	1 (17)	1 (11)	1.000
Carditis	n = 4	n = 9	
Median symptom duration (IQR), days	2 (1.5–14.5)	2 (2–10)	0.808
Any consultation with a cardiologist ^3^	4 (100)	9 (100)	
Median time to consultation with a cardiologist (IQR), days	0 (0–0.5)	0 (0–1)	0.604
Hospitalization	4 (100)	9 (100)	
Median length of hospital stay (IQR), days	6 (5–7)	12 (5–13)	0.436
Inpatient consultation with a cardiologist	4 (100)	9 (100)	
Follow-up with a cardiologist after discharge	2 (50)	6 (67)	1.000
Median time to follow-up with a cardiologist (IQR), days	22 (6–38)	40 (5–75)	1.000
Myocardial biopsy	4 (100)	8 (89)	1.000
Median time from carditis diagnosis to biopsy (IQR), days	1	4 (1–7)	0.109
Symptom improvement	2 (50)	8 (89)	0.203
Median duration of steroid use (IQR), days	42.5 (36–44)	39.5 (36–40)	0.368
Need for >1 steroid tapering course	1 (25)	1 (11)	0.491
Selective immunosuppressive therapy ^6^	1 (25)	3 (33)	1.000
Carditis response/remission at final follow-up ^5^	2 (50)	8 (89)	0.203
Hospital readmission	1 (25)	1 (11)	0.202
Recurrence	2 (50)	1 (11)	0.455
Mortality	3 (75)	2 (22)	0.213

^1^ Total nephritis cases within the 3-month study window: 2019 = 13; 2021 = 25. Total carditis cases: 2019 = 6; 2021 = 23. Only 2 patients developed symptoms/lab abnormalities indicating carditis prior to hospitalization (both in 2021). Neither received outpatient treatment, and only 1 was able to see a cardiologist before hospitalization. ^2^ In total, 8 patients presented with nephritis prior to hospitalization (4 in 2019 and 4 in 2021). The remaining patients all presented at the emergency department or hospital. ^3^ Includes as an outpatient before hospitalization, as an inpatient, or as an outpatient after hospital discharge. ^4^ Selective immunosuppressive therapy included infliximab, mycophenolate, azathioprine, or cyclophosphamide. ^5^ Defined as symptom improvement to Common Terminology Criteria for Adverse Events 5.0 grade 1 or below or normalization of lab abnormalities. ^6^ Selective immunosuppressive therapy included infliximab, anti-thymocyte globulin, mycophenolate, or abatacept.

## Data Availability

The datasets used and analyzed in this study are available from the corresponding author on reasonable request.

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
