# Peer review of "Practice Changes in Checkpoint Inhibitor-Induced Immune-Related Adverse Event Management at a Tertiary Care Center"

_cancers, 2024, doi:10.3390/cancers16020369_

Round 1

Reviewer 1 Report

Comments and Suggestions for Authors

Shatila and co-workers report in the present manuscript the comparison of the management of the immune-related adverse events in patients treated with immune checkpoint inhibitors at the MD Anderson Cancer Center in Houston, per- and post-COVID pandemic pediod. 

As these CPIs are more and more widely used, it is of utmost importance to improve the management of the adverse events relate to their use (and usually related to the response of patients to the disease). 

The design and the results are clearly exposed and explained, but maybe the very first paragraph of the Results (see comments below). 

The authors recognize the limitations of the present study that undoubtedly calls for a larger, multi-centered, prospective study. 

But no doubt the present manuscript is setting the solid basis for getting the larger one appropriately designed and funded. 

Few comments about the manuscript: 

L147 : most the common => the most common

Results, table 1 and figure 1.

The inter-relation between the paragraphs 3.1&3.2, the table 1 and the figure 1 is not intuitive.

In the figure 1, you report analysis of 732 patients with supposed irAEs from which you eventually excluded 245 patients. 487 patients were included in analysis. Among those, you report 79 and 161 cases of GI 240 cases of GI irAEs, while in the paragraph 3.1, you talk about many more (571 and 718 GI irAEs in 2019 and 2021 respectively). Should we understand these are numbers for the whole years and not only for the 3-month period from each year you focused on?

L365 : in diagnosis pneumonitis => in diagnosis of pneumonitis / diagnosing pneumonitis

Conclusion : I’d like the conclusion to be a little less definitive, e.g.

The current […] irAEs. Changes in practice, picked out in our study, between […] pandemic.

Although […] recent years, according to the irEA panel analyzed here.

We anticipate that Ensuring adequate […] disease recurrence, but this will need to be confirmed in a prospective, randomized controlled trial.  

Some of the reported p-values for statistical analyses seem to me counter-intuitive.

Could those be double-checked?

Especially :

-          L179 : , clinical 178 remission rates (92% in 2019 compared with 89% in 2021, P = 0.272)

-          Table 4 : Median symptom duration (IQR), days

-          Table s4 : Median symptom duration (IQR), days

-          Table s7 : Early consultation with a pulmonologist

-          Supp figure 4b : renal and cardiac group p-values

Author Response

Reviewer 1:

Comments and Suggestions for Authors

Shatila and co-workers report in the present manuscript the comparison of the management of the immune-related adverse events in patients treated with immune checkpoint inhibitors at the MD Anderson Cancer Center in Houston, per- and post-COVID pandemic period. 

As these CPIs are more and more widely used, it is of utmost importance to improve the management of the adverse events relate to their use (and usually related to the response of patients to the disease). 

The design and the results are clearly exposed and explained, but maybe the very first paragraph of the Results (see comments below). 

The authors recognize the limitations of the present study that undoubtedly calls for a larger, multi-centered, prospective study. 

But no doubt the present manuscript is setting the solid basis for getting the larger one appropriately designed and funded. 

 Thank you very much for your comments.

Few comments about the manuscript: 

L147 : most the common => the most common

Revised as suggested.

Results, table 1 and figure 1.

The inter-relation between the paragraphs 3.1&3.2, the table 1 and the figure 1 is not intuitive.

In the figure 1, you report analysis of 732 patients with supposed irAEs from which you eventually excluded 245 patients. 487 patients were included in analysis. Among those, you report 79 and 161 cases of GI 240 cases of GI irAEs, while in the paragraph 3.1, you talk about many more (571 and 718 GI irAEs in 2019 and 2021 respectively). Should we understand these are numbers for the whole years and not only for the 3-month period from each year you focused on?

Thank you for pointing this out. We confirm that reviewer’s understanding is correct that 3.1 refers to across the entire year while 3.2 and Figure 1 refers to within the designated 3 months period. We have added some text in lines 146 and 153 to clarify this.

L365 : in diagnosis pneumonitis => in diagnosis of pneumonitis / diagnosing pneumonitis

Revised as suggested.

Conclusion: I’d like the conclusion to be a little less definitive, e.g.

The current […] irAEs. Changes in practice, picked out in our study, between […] pandemic.

Revised as suggested. “The changes in practice between 2019 and 2021 we focused on […] pandemic.”

Although […] recent years, according to the irEA panel analyzed here.

Revised as suggested. “Although overall irAE outcomes among the toxicities we described […]”

We anticipate that Ensuring adequate […] disease recurrence, but this will need to be confirmed in a prospective, randomized controlled trial.  

Revised as suggested. Ensuring adequate […] disease recurrence, though prospective studies are needed to confirm these findings.”

Some of the reported p-values for statistical analyses seem to me counter-intuitive.

Could those be double-checked?

Thank you very much for pointing this out. We have double-checked the p-values and made corrections accordingly.

Especially :

-          L179: , clinical 178 remission rates (92% in 2019 compared with 89% in 2021, P = 0.272)

Thank you again for pointing this out. We double checked and corrected the p-value which is supposed to be 0.711. We have updated the table and manuscript accordingly.

-          Table 4 : Median symptom duration (IQR), days

This p-value is correct. Although it seems to be a major difference (77 days vs. 19 days), however, due to the small sample size (n=6 and 9 respectively not accounting for missing values). This limits the power of the analysis to detect significant differences.

-          Table s4 : Median symptom duration (IQR), days

Thank you for pointing this out. We have updated the supplementary table and manuscript accordingly.

-          Table s7 : Early consultation with a pulmonologist

Thank you very much for pointing this out. We confirm the p-value was correct.

-          Supp figure 4b : renal and cardiac group p-values

Same explanation as above regarding small sample sizes limiting statistical power despite what appear to be large differences.

Reviewer 2 Report

Comments and Suggestions for Authors

1.       Line 2: Add to title, Practice Changes in “Checkpoint Inhibitor Induced” Immune…

2.       The immune assessment here is based on checkpoint inhibitor therapy. Not all immune therapeutics induce the same toxic events. Grouping all checkpoint inhibitor toxic events as immune is misleading.

3.       Line25: Add understanding of “checkpoint inhibitor” (see 44-57) immune…

4.       Lines 56-57, …immunotherapy [4]”: Future immune therapy does not necessarily only involve checkpoint inhibitors. This is only reflective of CPI therapy not all immune therapy.

5.       Lines 58-59: This is true however, the clinical focus priority is not to address all AE’s related to CPI’s, it is to understand AE’s that are associated with fatality and reduce fatality. Fatal events associated with CPI’s occur in 0.3-1.3% of patients (Wang, Salem et al JAMA Oncol 2018), 70% from colitis with CTLA-4 inhibition and 35% pneumonitis, 22% hepatitis and 15% neurotoxic events from anti PD-1/PD-L1 inhibitors. Combination CTLA-4/PD-1/PD-L1 inhibitor therapy had a higher proportion of cardi toxic deaths (25% from myocarditis). These events are captured by this analysis.

6.       Lines 76-78, “…irAE management…patient outcomes”: Great value.

7.       Line 85, “…high mortality…”: See above comments and expand discussions in introduction to this effect.

8.       Line 163, Figure 1: Is there any assessment on neurotoxicity? What about duration of toxicity while on treatment with CPI? Length of delay to addressing toxic event is related as well.

9.       Lines 171-172:  If not statistically significant then would not make conclusive statement like “were lower”, would say “suggestive of” or something similar.

10.   Lines 180-183, “…with significant…P=0.014)”: Very important! Would tone down non-significant discussion and increase focus of 180-183 parameters!

11.   Lines 221-233: Would control use of inappropriate conclusive statements in setting with non significant results. This deters from the impressive statistically significant results. Further assessment of trend or suggestive results can be further studied. However, the conclusive results of this work should be the main focus to highlight.

12.   Lines 279-283, “although…P=0.202”: So this statement proven by your data is false. P is > 0.05. Would state such and reduce discussion. Would also add there is suggestion of difference based on numbers and further assessment with larger number of patients will be required.

13.   Line 286: Lack of assessment of neurotoxic events is a big limit. Would suggest attempt to somehow capture neurotoxic events.

14.   Lines 290-295, “we found…attempts”: Would go through statistically significant conclusive results and separate for non significant results. Would conclude factors associated with significance and consider non significant results that are suggestive as possibly important to study further as they may also be related but with current results are not conclusive.

15.   Line 296, “…importance…algorithm…”: Agree, but only based on the conclusive results.

16.   Line 324, Figure 2: So Figure 2 is a very good concept but I don’t see a single defined practice/facility/physician change that is not occurring in 2019 and is occurring in 2021. Typically, there are physician guidelines (i.e. at onset of Grade 2 diarrhea initiate a steroid dose level within 24 hours), patient to clinic required visit guidelines (i.e. temperature > 101.5), endoscopy/colonoscopy guidelines (i.e. relate to level of diarrhea), EKG guidelines [prophylactic EKG’s (i.e. based on chest pain, fatigue…] and hospital admission guidelines.

17.   Line 340, “Pulmonary irAEs…”: Exactly. So what measurable guidelines are done (i.e. routine O2 assessment at clinic visit, pulmonary function studies, imaging of chest…)? And how was this altered from 2019 to 2021? This comes across as “we are better but don’t know why”. Please explain why so that others who read this can also learn and enhance control measures for patient management while taking CPI’s.

Author Response

Comments and Suggestions for Authors

  1. Line 2: Add to title, Practice Changes in “Checkpoint Inhibitor Induced” Immune…

Revised as suggested.

  1. The immune assessment here is based on checkpoint inhibitor therapy. Not all immune therapeutics induce the same toxic events. Grouping all checkpoint inhibitor toxic events as immune is misleading.

Thank you for your comment. We revised the introduction to clarify it as best as we can.

  1. Line25: Add understanding of “checkpoint inhibitor” (see 44-57) immune…

Added to lines 26-27.

  1. Lines 56-57, …immunotherapy [4]”: Future immune therapy does not necessarily only involve checkpoint inhibitors. This is only reflective of CPI therapy not all immune therapy.

Revised as suggested.

  1. Lines 58-59: This is true however, the clinical focus priority is not to address all AE’s related to CPI’s, it is to understand AE’s that are associated with fatality and reduce fatality. Fatal events associated with CPI’s occur in 0.3-1.3% of patients (Wang, Salem et al JAMA Oncol 2018), 70% from colitis with CTLA-4 inhibition and 35% pneumonitis, 22% hepatitis and 15% neurotoxic events from anti PD-1/PD-L1 inhibitors. Combination CTLA-4/PD-1/PD-L1 inhibitor therapy had a higher proportion of cardi toxic deaths (25% from myocarditis). These events are captured by this analysis.

We agree with reviewer’s comments and have expanded our introduction to accommodate this with the following statement: “Mortality rates have been estimated at 0.3-1.3% with colitis, pneumonitis, carditis, hepatitis, and neurotoxicity among the most fatal toxicities”.

  1. Lines 76-78, “…irAE management…patient outcomes”: Great value.

Thank you.

  1. Line 85, “…high mortality…”: See above comments and expand discussions in introduction to this effect.

Revised as suggested.

  1. Line 163, Figure 1: Is there any assessment on neurotoxicity? What about duration of toxicity while on treatment with CPI? Length of delay to addressing toxic event is related as well.

We did not include neurotoxicity as it is difficult for IT to pull these cases with straightforward parameters, like we used for the toxicities we included. A high demand on manual chart review will be required for screening and confirmation which is not feasible for us. We have added this to the limitation.

We have reported variables on symptom duration as well as time to initial outpatient treatment and time to steroids/consultation in our analyses to try to account for what reviewer mentioned in the second and third statement.

  1. Lines 171-172:  If not statistically significant then would not make conclusive statement like “were lower”, would say “suggestive of” or something similar.

Revised as suggested.

  1. Lines 180-183, “…with significant…P=0.014)”: Very important! Would tone down non-significant discussion and increase focus of 180-183 parameters!

Revised as suggested.

  1. Lines 221-233: Would control use of inappropriate conclusive statements in setting with non significant results. This deters from the impressive statistically significant results. Further assessment of trend or suggestive results can be further studied. However, the conclusive results of this work should be the main focus to highlight.

Revised as suggested.

  1. Lines 279-283, “although…P=0.202”: So this statement proven by your data is false. P is > 0.05. Would state such and reduce discussion. Would also add there is suggestion of difference based on numbers and further assessment with larger number of patients will be required.

Revised as your recommendations.

  1. Line 286: Lack of assessment of neurotoxic events is a big limit. Would suggest attempt to somehow capture neurotoxic events.

We will not be able to capture neurotoxic events in this study as the response #8 stated, but added to the limitations: “There would be merit in expanding the sample size and including irAEs from other organ systems such as cutaneous or neurological irAEs”.

  1. Lines 290-295, “we found…attempts”: Would go through statistically significant conclusive results and separate for non significant results. Would conclude factors associated with significance and consider non significant results that are suggestive as possibly important to study further as they may also be related but with current results are not conclusive.

Thank you for your input. The authors feel we have mentioned the main significant results here without it being a rehash of the results section. This paragraph serves to introduce the discussion and we focused on the few results (procedural evaluation rates, post-discharge follow-up, timely consultation) that were particularly important to the clinical practice.

  1. Line 296, “…importance…algorithm…”: Agree, but only based on the conclusive results.

We agree with this statement.

  1. Line 324, Figure 2: So Figure 2 is a very good concept but I don’t see a single defined practice/facility/physician change that is not occurring in 2019 and is occurring in 2021. Typically, there are physician guidelines (i.e. at onset of Grade 2 diarrhea initiate a steroid dose level within 24 hours), patient to clinic required visit guidelines (i.e. temperature > 101.5), endoscopy/colonoscopy guidelines (i.e. relate to level of diarrhea), EKG guidelines [prophylactic EKG’s (i.e. based on chest pain, fatigue…] and hospital admission guidelines.

Our institution does have guidelines in place for all toxicities and there are multiple society guidelines for checkpoint inhibitor toxicities. MD Anderson ICI Colitis guideline link is https://www.mdanderson.org/content/dam/mdanderson/documents/for-physicians/algorithms/clinical-management/clin-management-immune-mediated-colitis-web-algorithm.pdf. The guideline for renal toxicity is https://mdandersonorg.sharepoint.com/teams/Immuno-oncologicToxicityWorkingGroup/Shared%20Documents/IOTOX%20Guidelines/2.%20clin-management-nephritis-web-algorithm%20(version%201%20revised).pdf?csf=1&web=1&e=2JpR7u. The cardiac and pulmonary toxicity guidelines are not approved in public domain, and currently available to MD Anderson internal use only. Unfortunately, the development of these guidelines does not always translate the strict compliance. What we present in Figure 2 is a culmination of all possible factors that may lead to our significant findings including but not limited to COVID, study design, oncology team familiarity with these adverse events, and hospital related factors including non-compliance to the algorithms/guidelines that we are discussing and an overburdening of the specialty service.

  1. Line 340, “Pulmonary irAEs…”: Exactly. So what measurable guidelines are done (i.e. routine O2 assessment at clinic visit, pulmonary function studies, imaging of chest…)? And how was this altered from 2019 to 2021? This comes across as “we are better but don’t know why”. Please explain why so that others who read this can also learn and enhance control measures for patient management while taking CPI’s.

We believe one of the main reasons for our findings was better education across the institution regarding these adverse events which may improve how compliant the oncology team follows the institutional guideline, and how frequent the primary team is to consult specialty teams. Added a line on this in the discussion: “This is likely due to an increased institutional effort to host educational events to raise awareness regarding these toxicities”. While we agree that the why of it is important, we are also interested in showing more on how we have changed in this 2-year period that may have impacted our quality of care.

Reviewer 3 Report

Comments and Suggestions for Authors

1-  An intense yet excellent presentation exploring Immune-related adverse events, with the intention of improving management and outcome of patients with gastrointestinal disease.

2-  This is an area which is increasing in importance for various other diseases. 

3-  I have seen few manuscripts, with such attention to detail and presentation to explain the adverse events, and use the data to improve patient management. This is well indicated in the tables and figures presented.

4-  As mentioned, this is an area which which is gaining a great deal of importance in the management of patients with various diseases, as evident in the number of publications  recently published.

5-  I also find it gracious that the authors are willing to provide additional data if requested by others in this field.

Author Response

Comments and Suggestions for Authors

Thank you so much for your very kind and encouraging words. We have tried our best to capture a screenshot of a very important and dense topic.

1-  An intense yet excellent presentation exploring Immune-related adverse events, with the intention of improving management and outcome of patients with gastrointestinal disease.

2-  This is an area which is increasing in importance for various other diseases. 

3-  I have seen few manuscripts, with such attention to detail and presentation to explain the adverse events and use the data to improve patient management. This is well indicated in the tables and figures presented.

4-  As mentioned, this is an area which which is gaining a great deal of importance in the management of patients with various diseases, as evident in the number of publications  recently published.

5-  I also find it gracious that the authors are willing to provide additional data if requested by others in this field.

Round 2

Reviewer 2 Report

Comments and Suggestions for Authors

The response by authors is all good.